# The regulation of a pigmentation gene in the formation of complex color patterns in *Drosophila* abdomens

**Komal K. B. Raja**[1], **Mujeeb O. Shittu**[2], **Peter M. E. Nouhan**[3], **Tessa E. Steenwinkel**[4], **Evan A. Bachman**[5], **Prajakta P. Kokate**[4], **Alexander McQueeney**[6], **Elizabeth A. Mundell**[7], **Alexandri A. Armentrout**[4], **Amber Nugent**[4], **Thomas Werner**[4]*

1 Department of Pathology & Immunology, Baylor College of Medicine, Houston, Texas, United States of America, 2 Department of Biotechnical and Clinical Laboratory Science, Jacobs School of Medicine and Biomedical Science, University at Buffalo, The State University of New York (SUNY), New York, United States of America, 3 McCourt School of Public Policy, Georgetown University, Washington, D.C., United States of America, 4 Department of Biological Sciences, Michigan Technological University, Houghton, Michigan, United States of America, 5 Michigan State University, College of Human Medicine, East Lansing, Michigan, United States of America, 6 School of Medicine, Eberhard Karls University of Tübingen, Geschwister-Scholl-Platz, Tübingen, Germany, 7 School of Technology, Michigan Technological University, Houghton, Michigan, United States of America

* twerner@mtu.edu

**Data Availability Statement:** All relevant data are within the paper and its Supporting Information files.

## Abstract

Changes in the control of developmental gene expression patterns have been implicated in the evolution of animal morphology. However, the genetic mechanisms underlying complex morphological traits remain largely unknown. Here we investigated the molecular mechanisms that induce the pigmentation gene *yellow* in a complex color pattern on the abdomen of *Drosophila guttifera*. We show that at least five developmental genes may collectively activate one *cis*-regulatory module of *yellow* in distinct spot rows and a dark shade to assemble the complete abdominal pigment pattern of *Drosophila guttifera*. One of these genes, *wingless*, may play a conserved role in the early phase of spot pattern development in several species of the quinaria group. Our findings shed light on the evolution of complex animal color patterns through modular changes of gene expression patterns.

## Introduction

Pigmentation patterns are among the most beautiful and striking phenotypic features displayed by animals. They are known to play important roles in adaptation (camouflage and mimicry), reproduction (mate choice), and physiology (UV protection and thermoregulation) [1, 2]. Melanin pigments are commonly found in both invertebrates and vertebrates. The biochemical pathway that leads to melanin synthesis is well understood. Several studies have reported that the *yellow* (*y*) gene is required for the formation of black melanin in insects [3–9], where it plays a pertinent role in the evolution and diversification of pigmentation patterns. Spatio-temporal changes of *y* gene expression due to mutations in *y* cis-regulatory modules (CRMs) as well as changes in the deployment of transcription factors that control *y* CRMs

**Funding:** This work was funded to Thomas Werner by the National Institutes of Health, General Medical Sciences, grant number 1R15GM107801-01A1. https://www.nih.gov/ The funders had no role in study design, data collection and analysis, decision to publish, or preparation of the manuscript.

**Competing interests:** The authors have declared that no competing interests exist.

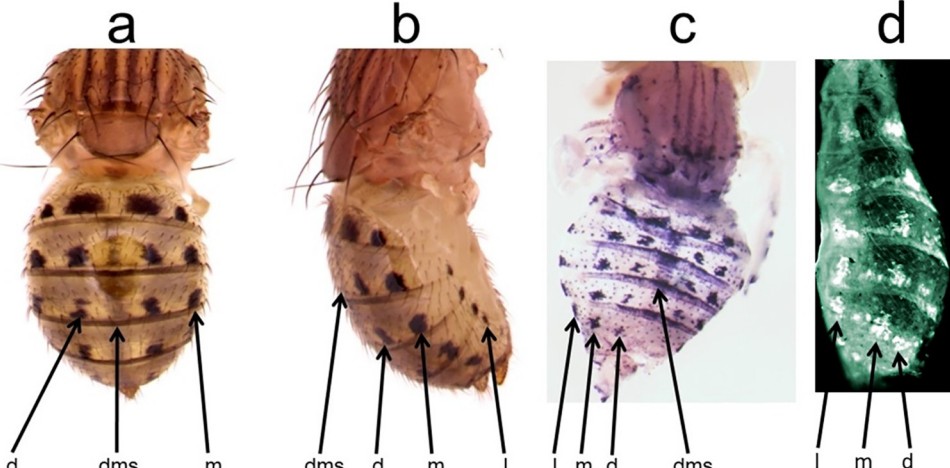

**Fig 1. The *y* gene expression pattern prefigures the adult pigment pattern of *D. guttifera*.** a, Adult, dorsal view showing the dorsal midline shade and the dorsal and median spot rows. b, Adult, lateral view. All three spot rows are visible along with the dorsal midline shade c, *yellow* mRNA expression pattern in a pupal abdomen at stage P10 showing three pairs of longitudinal spots and a dorsal midline shade. d, Yellow protein expression pattern in a P11 pupal abdomen. dms = dorsal midline shade, d = dorsal, m = median, l = lateral spot rows.

have contributed to the pigmentation diversity that we see among *Drosophila* species [3, 10–15]. Prior studies have identified *y* CRMs and some of the *trans*-factors (i.e., Abdominal-A (Abd-A), Abdominal-B (Abd-B), Engrailed (En), and Distal-less (Dll)) that regulate *y* to establish simple pigment patterns on the wings and abdomens of *Drosophila* species [4, 12, 16–18]. However, how complex color patterns are encoded in DNA, generated during development, and evolve over time, is poorly understood.

To better understand the development and evolution of complex color patterns, we investigated the great variety of abdominal color patterns displayed by members of the quinaria species group within the genus *Drosophila*, which started diverging ~10 million years ago (MYA), while the common ancestor of *Drosophila melanogaster* and the quinaria group was dated back to 60 MYA [8, 19–23]. We primarily focused on *Drosophila guttifera* (*D. guttifera*), which displays the most complex abdominal color pattern of this group, consisting of four sub-patterns: a dorsal, a median, and a lateral pair of spot rows, as well as a dorsal midline shade (Fig 1A and 1B) [8, 19, 21]. Most other species of the quinaria group lack at least one of the four sub-patterns, providing an intriguing example of pattern modularity among species and thus a good foundation for studying the changes underlying complex pattern formation.

In this study, we show that only one CRM of *y* controls the entire abdominal spot pattern in *D. guttifera*. Furthermore, the developmental gene *wingless* (*wg*) appears to be the upstream inducer of the spotted pattern. In several members of the quinaria group, both *wg* and *y* expression patterns strongly correlate with the spot pattern diversity displayed by this species group.

## Results

### Gene expression patterns prefigure diverse abdominal spot patterns in species of the quinaria group

We first determined whether the *y* gene is expressed in developing pupae where black pigment will form on the adult abdomen of *D. guttifera*. Using *in situ* hybridization (ISH) and immuno-histochemistry in *D. guttifera* pupae, we found that the *y* mRNA and Y protein expression

patterns accurately prefigure the complex spot and shade pattern on adult flies (Fig 1). In order to identify putative upstream activators of *y*, we performed an ISH screen for 110 developmental genes (that were likely or proven to play a role in melanin formation as toolkit, immune, or terminal pigmentation genes) to detect those whose expression patterns prefigure that of the *y* gene (S1 Table) [24]. We found that *wg* expression precisely foreshadows the six rows of black spots (Fig 2B and S1–S4 Figs). Additionally, *decapentaplegic* (*dpp*) expression foreshadowed the dorsal and median pairs of spot rows (Fig 2C and S5A Fig), while *abdominal-A* (*abd-A*) expression correlated with the lateral pair of spot rows and the dorsal midline shade (Fig 2D, 2E and S5B Fig). Additionally, *hedgehog* (*hh*) and *zerknullt* (*zen*) were expressed along the dorsal midline of the abdomen (Fig 2F and 2G, S5C and S5D Fig). Thus, the activation of the *D. guttifera* color pattern appears to be induced in a modular fashion, which agrees with our observation that abdominal pigmentation patterns within the quinaria group are variations of the *D. guttifera* pattern ground plan (Fig 3). This situation is reminiscent of the wing pattern ground plan in nymphalid butterflies [25, 26]. Some examples of genes that did not show any detectable spot or shade pattern are shown in S6 and S7 Figs.

We next asked whether the abdominal spot pattern of a species closely related to *D. guttifera* shares a similar gene expression pattern modularity [22]. We thus performed ISH experiments in *Drosophila deflecta* (*D. deflecta*), which displays six longitudinal spot rows on its abdomen but, notably, lacks the dorsal midline shade (Fig 4A and 4B). As in *D. guttifera*, *y* mRNA in *D. deflecta* pupal abdomens was expressed in six rows of spots, but not along the dorsal midline, as we expected (Fig 4C–4E). Similarly, *wg* foreshadowed all six rows of spots, while *dpp* expression matched all but the lateral spot rows, just like in *D. guttifera* (Fig 5B and 5C, S8A Fig). In contrast to the *D. guttifera* results, *abd-A*, *hh*, and *zen* were absent along the dorsal midline, which agrees with the lack of dark pigment in *D. deflecta* adults (Fig 5D–5G and S8B). However, *abd-A* expression was not detectable where the lateral spot rows will form (Fig 5D), suggesting that these particular spots may be controlled by a different gene in *D. deflecta*.

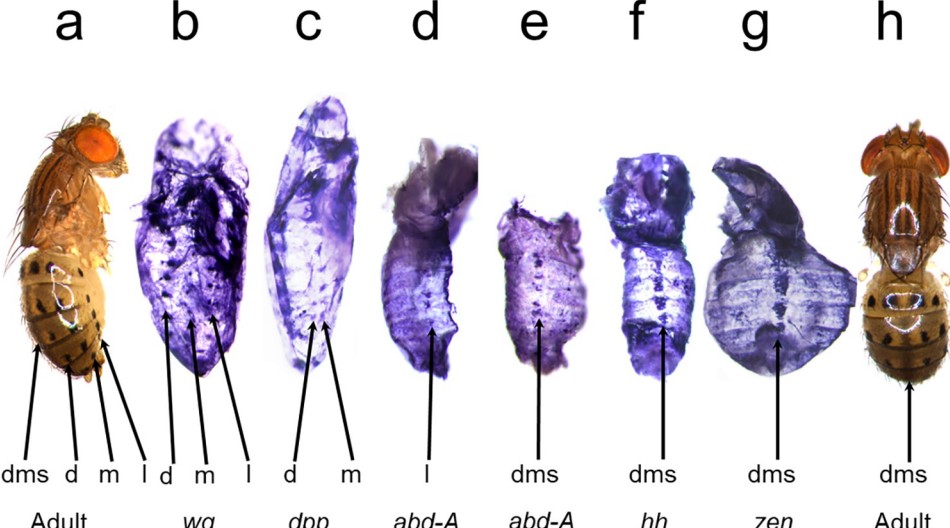

**Fig 2. The mRNA expression patterns of five developmental genes foreshadow the *y* expression pattern in *D. guttifera*.** a, Adult, lateral view. b-g, ISH in pupal abdomens. b-d, show lateral orientation. e-g, dorsal orientation of the pupal abdomen. b, *wg* expression is detected in all abdominal spot rows at pupal stage P7. c, P7 pupal abdomen showing *dpp* expression in the dorsal and median spot rows. d, P9 pupal abdomen showing *abd-A* in the lateral spot row. e, P9 pupal abdomen showing staining of an *abd-A* probe along the dorsal midline. f-g, *hh* and *zen* expression patterns were detected in P10 pupal abdomens. ISH was performed on pupae at stages P6 –P10 to screen for the mRNA transcripts of each gene. h, Adult, dorsal view. dms = dorsal midline shade, d = dorsal, m = median, l = lateral spot rows.

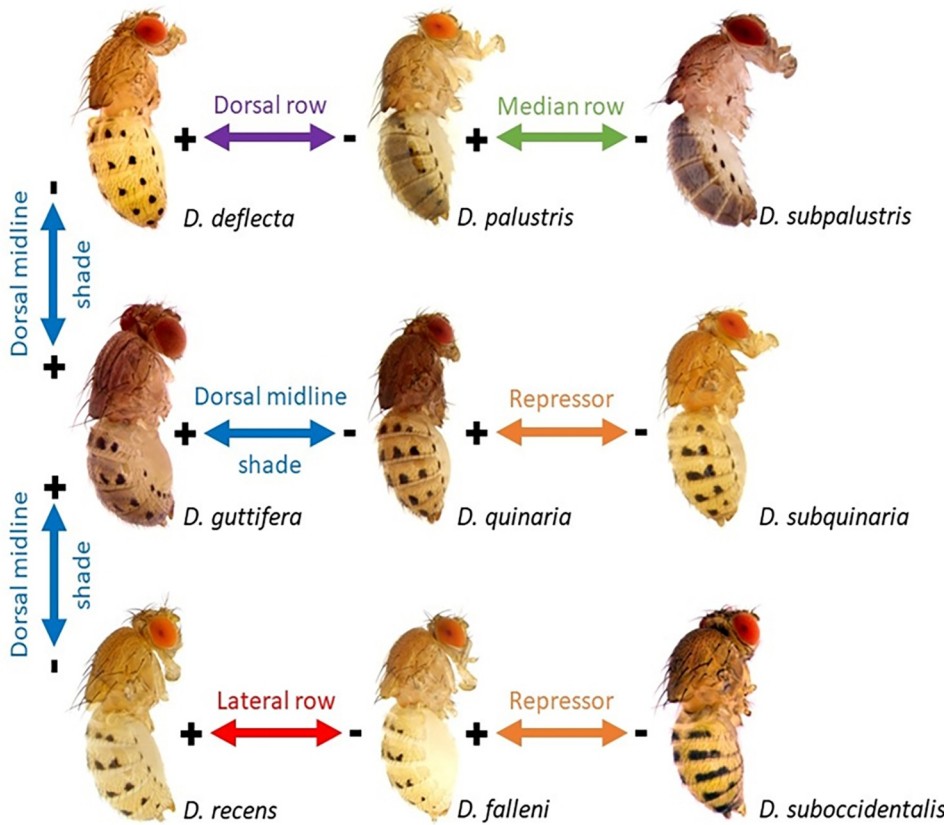

**Fig 3. Deviations from the *D. guttifera* ground plan led to the abdominal color pattern diversity in the quinaria species group.** + = gain, - = loss of a pattern element. "Repressor" suggests that stripes may be broken into spots by unidentified repressors of pigmentation. The arrows solely illustrate the modularity of these complex patterns and do not imply any evolutionary direction.

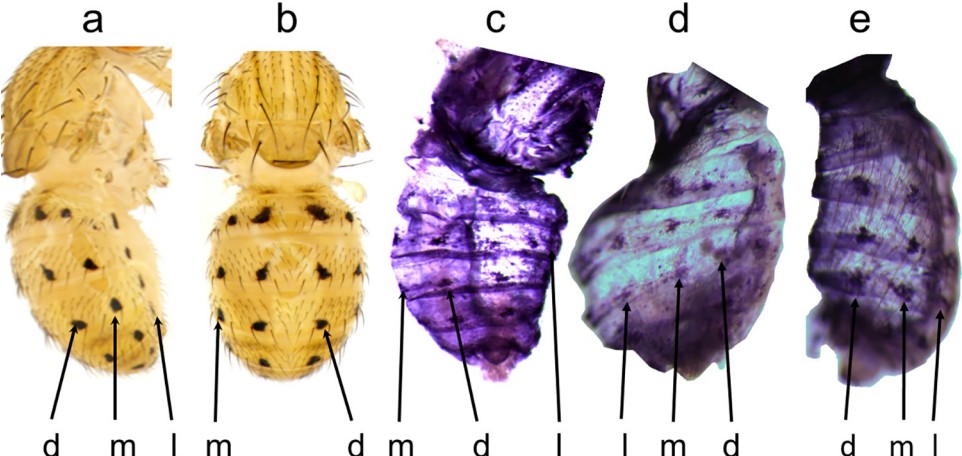

**Fig 4. The *y* gene expression pattern in *D. deflecta* foreshadows the black spot pattern on the adult abdomen.** a, Adult lateral view. b, Adult dorsal view. c-e, *y* mRNA in the P10 pupal epidermis prefigures the adult spot pattern. d = dorsal, m = median, l = lateral.

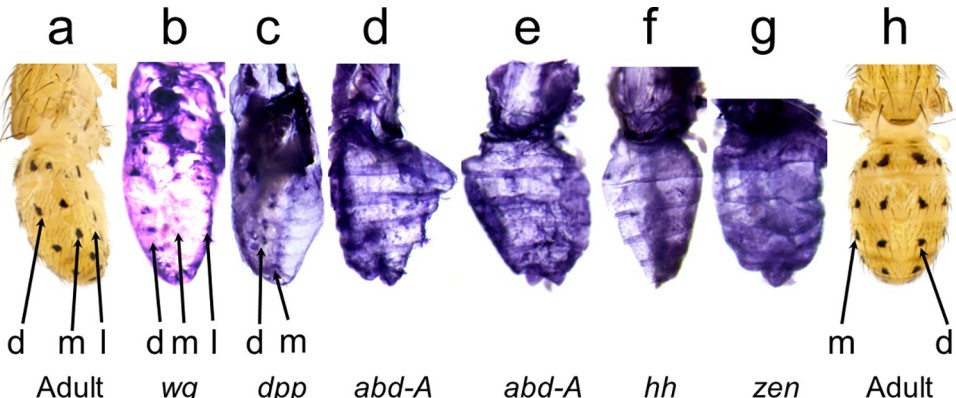

**Fig 5. The mRNA expression patterns of *wg* and *dpp* show correlation with *y* expression in adult *D. deflecta*.** a, h, Adult *D. deflecta* lateral and dorsal view. b, c, Lateral view. Developmental gene expression patterns of *wg* and *dpp* at pupal stage P7 in *D. deflecta* foreshadow distinct subsets of the adult abdominal color pattern. d-g, Dorsal view. *abd-A*, *hh*, and *zen* are not expressed along the dorsal midline of P9 and P10 pupae. ISH was performed on pupae at stages P6 –P10 to screen for the mRNA transcripts of each gene.

We next tested if the developmental genes that prefigured the adult color patterns show different expression patterns in two other closely related species that have fewer spot rows than *D. guttifera* and *D. deflecta*. Out of the five developmental genes that are expressed in a matching prepattern, we focused on the *wg* gene because its expression pattern foreshadowed all six spot rows in both *D. deflecta* and *D. guttifera*, while *wg* is also known to induce the black spots on the wings of *D. guttifera* [6]. To strengthen the evidence between the *wg* expression pattern in developing abdomens and the black spots on the adult abdomens in *D. guttifera* and *D. deflecta*, we tested if the correlation holds up even in species that display successively fewer rows of black spots—*Drosophila palustris* (*D. palustris*) and *Drosophila subpalustris* (*D. subpalustris*). These two quinaria group members display partial patterns of the *D. guttifera* ground plan, i.e., *D. palustris* and *D. subpalustris* lack two or four rows of spots, respectively (Fig 6A and 6D). Even

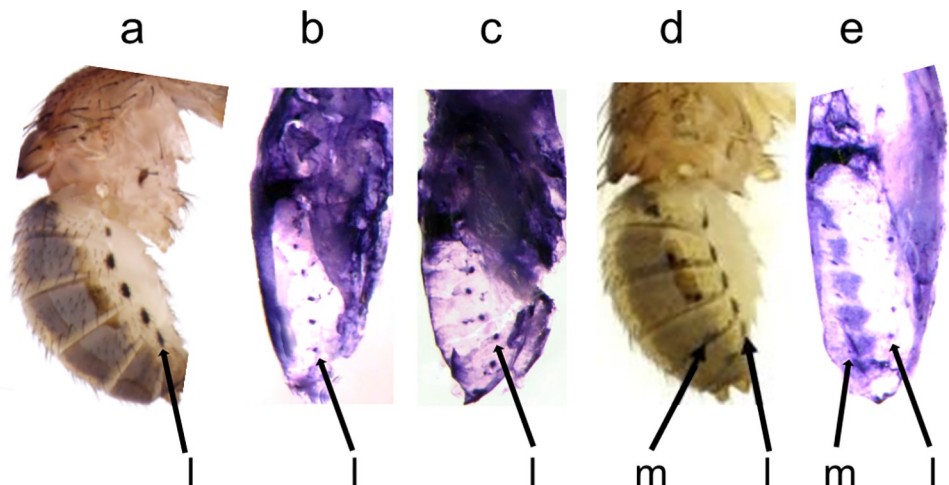

**Fig 6. Pupal *wg* expression in *D. palustris* and *D. subpalustris* precisely foreshadows the adult abdominal spot patterns.** a, Lateral view of an adult *D. subpalustris* showing a lateral spot row. b, c, *wg* mRNA expression during pupal stage P7 of *D. subpalustris*. *wg* expression is observed in the lateral spot row in two different pupal abdomens. d, Lateral view of an adult *D. palustris* showing lateral and median spot rows. e, *wg* mRNA expression during pupal stage P7 of *D. palustris*. The spot rows are designated as lateral (l) and median (m).

in these two additional species, our ISH results showed that *wg* was expressed only at sites that will develop black spots in the adult (Fig 6B, 6C and 6E), thus strengthening the correlation among four species with varying pigmentation patterns. We have shown earlier that *y* mRNA also precisely foreshadows the adult abdominal spot patterns of *D. palustris* and *D. subpalustris* [27], suggesting that *wg* induces the abdominal spot patterns by activating the *y* gene in several members of the quinaria group, like it is true for the *D. guttifera* wing spot pattern.

## The complex abdominal spot pattern of *D. guttifera* is reproduced by a single *y* CRM

Several *y* gene CRMs have been identified in various *Drosophila* species, and changes in these CRMs and/or in the deployment of *trans*-factors that regulate *y* gene expression have been implicated in the diversification of wing and body pigment patterns [6, 12–16]. We hypothesized that the developmental genes *wg*, *dpp*, *abd-A*, *hh*, and *zen* activate the *y* gene through four CRMs, each controlling one sub-pattern to assemble the complete melanin pattern on the abdomen. Hence, we searched for these CRMs by transforming *D. guttifera* with *DsRed* reporter constructs containing non-coding fragments of the 42 kb *D. guttifera y* gene locus (S9 Fig) [6]. Surprisingly, only one 953-bp fragment from the *y* intron, the *gut y spot* CRM, drove expression closely resembling all six spot rows and the dorsal midline shade on the developing pupal abdomen (Fig 7 and S10 Fig). To isolate possible sub-pattern-inducing CRMs, we first subdivided the *gut y spot* CRM into two partially overlapping sub-fragments. Unexpectedly, the 636-bp left sub-fragment displayed stripe expression parallel to the segment boundary (hereafter referred to as "horizontal stripe expression", when viewing the fly with the head on top and the abdomen on the bottom of the picture) along the posterior edges of each abdominal segment, while the 570-bp right fragment was inactive (#1 & #2, Fig 7). Further dissection

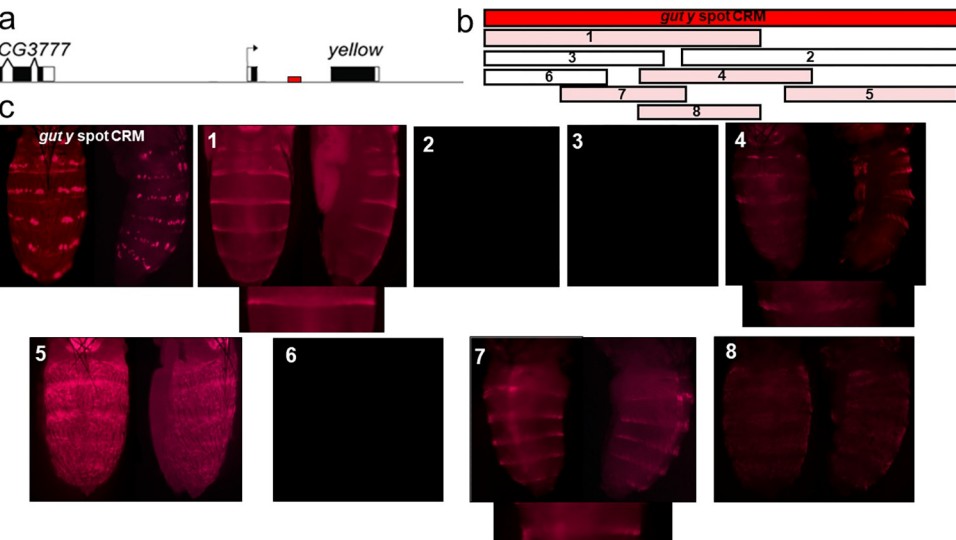

**Fig 7. The *gut y spot* CRM is harbored within the *y* intron.** a, The *y* gene locus. The red bar indicates the relative position of the *gut y spot* CRM. b, Subdividing the *gut y spot* CRM revealed horizontal stripes on each abdominal segment. The rectangular bars (1–8) represent sub-fragments of the *gut y spot* CRM and the corresponding pupal *DsRed* expression patterns in transgenic *D. guttifera* during stage P10 are shown in c. The sub-fragments that drove reporter expression are highlighted in pink (#1, #4, #5, #7, and #8), whereas the constructs that did not show any pattern are shown as rectangular white bars (#2, #3, and #7). For sub-fragments #1, #4 and #7, an additional image focusing on a single stripe is shown below the abdominal images that show horizontal stripes on the abdomen. Sub-fragment #7 shows the minimal *y stripe core* CRM with reporter expression along the dorsal midline.

of the left sub-fragment revealed a 259-bp sub-fragment, which contained the minimal *gut y core stripe* CRM with some additional dorsal midline shade activity (#7 Fig 7). We also dissected the right sub-fragment into smaller overlapping fragments (#4 and #5, Fig 7). We observed that a 405-bp sub-fragment #5 drove ubiquitous reporter expression on the abdomen, which could be an artifact resulting from dissecting the *y* spot regulatory element into smaller sub-fragments. It is well known that disrupting CRMs can lead to spurious expression patterns. Conversely, the entire abdomen of *D. guttifera* shows light pigmentation in comparison with *D. deflecta* (Figs 1–5). This sub-fragment #5 may be driving the dark ubiquitous abdominal pigmentation in *D. guttifera*. The corresponding orthologous fragment of *D. deflecta* (shown in Fig 8 #5) does not show any such pattern when transformed in *D. guttifera*.

Currently, we cannot offer any direct evidence for specific candidate repressor genes. Neither the ISH experiments nor the bioinformatics analysis, using JASPAR, resulted in putative pigment stripe repressors (S1 File, S11 Fig and S2 Table). Although we identified 24 Engrailed (En)-binding sites and 19 Homothorax (Hth)-binding sites in the *gut y spot* CRM (both are known repressors of pigmentation in *Drosophila*) [12, 17], these sites were not enriched in the right half of the CRM, as we would have expected (S11 Fig). However, our transcription factor binding site analysis of the *gut y spot* CRM sequence revealed putative binding sites for most of the developmental genes that we identified as potential activators in our ISH screen (S1 File and S11 Fig), except for *Cubitus interruptus* (*ci*, the gene encoding the transcription factor downstream of *hh*). This suggests that localized spot activation by some of our identified developmental factors contributes to the formation of the abdominal spot pattern. We also identified other tissue-specific CRMs within the *y* locus [3, 10]. We found the wing-body CRM approximately 4 kb upstream of the *y* transcription start site, whereas the CRMs that specify *y* expression in the head, thorax, bristles, mouth parts, legs, and trachea were found within the *y*

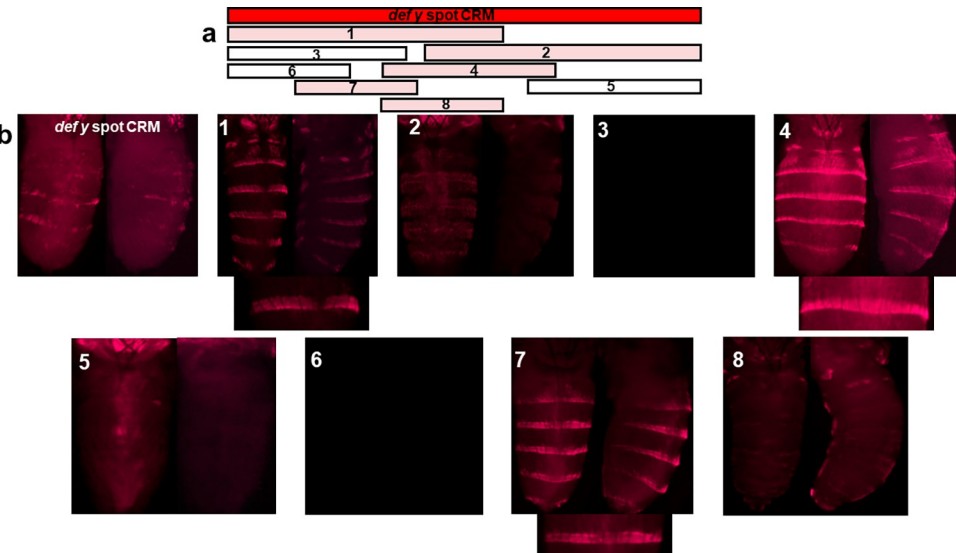

**Fig 8. The orthologous *D. deflecta* region (*def y spot* CRM) analyzed in transgenic *D. guttifera*.** a, A schematic of the orthologous *def y spot* CRM is shown as a solid red bar. The rectangular bars (1–8) under the red bar represent the sub-fragments of the *def y spot* CRM that were tested for reporter activity. The sub-fragments that drove reporter expression are highlighted in pink, whereas the constructs that did not show any pattern are shown as rectangular white bars. b, The pupal *DsRed* expression patterns of the sub-fragments are shown in transgenic *D. guttifera* during stage P10. For sub-fragments #1, #4 and #7, an additional image focusing on a single stripe is shown below the abdominal images that show horizontal stripes on the abdomen. Sub-fragment #7 shows the orthologous minimal *def y stripe core* CRM. The reporter expression is detected in horizontal stripes but absent along the dorsal midline.

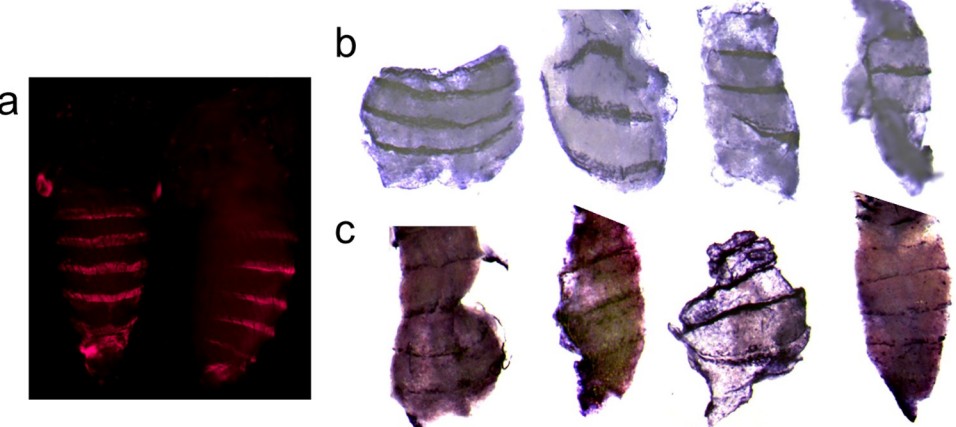

**Fig 9. ISH signals showing that the *hh* cDNA and *wg* cDNA ectopic expression patterns correlate with the *gut y* stripe CRE reporter assay.** a, DsRed reporter assay showing the activity of the *gut y* stripe CRE at stage P10. This construct drives reporter expression in horizontal stripes across the abdomen. b, ISH signal of the ectopic expression of *hh* cDNA along the tergites of P10 pupae of *D. guttifera* showing a stripe pattern. c, ISH signal showing the ectopic expression of *wg* cDNA along the tergites of P10 pupae of *D. guttifera*. The probe is detected in horizontal stripes. Negative controls without any significant abdominal staining are shown in S14 Fig.

intron. In addition, one 1573-bp fragment located 2 kb upstream of the *y* transcription start site, as shown in Fig 9A, drove reporter expression on the abdomen in horizontal stripes and resembles the reporter activity of the *gut y core stripe* CRM (Fig 7C). This fragment ("*gut y stripe* CRM") may contain a redundant CRM, and it is possible that the spot pattern output could be due to a combinatorial activity between the two stripe CRMs and the elusive repressors.

## The core stripe CRM is conserved among *D. guttifera* and *D. deflecta*

To understand how changes in CRMs and *trans*-acting factors play a role in the evolution of the *D. guttifera* spot pattern, we cloned the 938-bp orthologous abdominal spot CRM from *D. deflecta*, the *def y spot* CRM, and transformed it into *D. guttifera*, using the *DsRed* reporter assay. The *def y spot* CRM drove faint dorsal spot row and stripe expression, most strongly where the dorsal spots are located (Fig 8). We further subdivided the *def y spot* CRM into 8 sub-fragments and identified a minimal *def y core stripe* CRM (288 bp) (#7 Fig 8). This sub-fragment drove a striped pattern, but without the dorsal midline shade activity that we saw in the *D. guttifera* minimal *gut y core stripe* CRM (Fig 7 #7). To test if *D. melanogaster trans*-factors can activate these two spot CRMs, we further transformed the *gut y spot* and *def y spot* CRMs and all sub-fragments into *D. melanogaster*. As a result, none of the reporter constructs showed any expression in transgenic *D. melanogaster* pupal abdomens. As the spot CRMs from *D. guttifera* and *D. deflecta* are not orthologous to any sequences within the *D. melanogaster y* locus, changes in *cis* have contributed to the diversification of pigment patterns between *D. melanogaster* and the quinaria species group. The *y spot* CRMs of *D. guttifera* and *D. deflecta* show 74% sequence identity, and their sequence alignment is shown in S2 File. The remainder of the sequence has strongly diverged.

## Functional studies to provide evidence for the roles of *wg*, *dpp*, *hh*, and *abd-A* in pigmentation

Although the correlative evidence provided by the mRNA expression data and the transcription factor binding site analysis of the *gut y spot* CRM sequence are interesting and thought-

provoking, they do not provide conclusive evidence for the involvement of the developmental genes in the process of pigmentation. To investigate these correlations further, we performed functional sufficiency experiments by ectopically expressing the cDNAs of *wg*, *hh*, and *abd-A* in transgenic *D. guttifera* by using the *gut y* stripe CRM as a driver. The *gut y* stripe CRM drives expression in the tergites along the intersegmental regions on the *D. guttifera* abdomen at stage P10 (Fig 9A). All three cDNAs were commercially available and derived from *D. melanogaster*. Our aim was to change the spot pattern on the abdomen of *D. guttifera* into stripes. As a result, the adult pigment pattern did not change into stripes. There are at least five possible explanations: 1) The driver does not work, and the developmental genes are not expressed at all. 2) The driver works, but the developmental genes are not sufficient to cause ectopic pigmentation. 3) The developmental genes are sufficient, but the driver acts too late to cause ectopic pigmentation. 4) Multiple developmental genes need to be simultaneously expressed to achieve the desired ectopic expression. 5) The presumptive endogenous repressor(s) is strong enough to suppress the function of the ectopically expressed genes. To investigate this issue further, we tested if and when the *gut y* stripe CRM drives the *D. melanogaster*-specific *wg*, *hh*, and *abd-A* transcripts in the transgenic *D. guttifera* pupae by performing ISH experiments at a variety of pupal stages (P7 –P12). We saw that the *wg* and *hh* cDNAs are indeed ectopically expressed in stripes on the tergites around the intersegmental regions of P10 pupae (Fig 9B and 9C), while *abd-A* did not show any expression, confirming that our overexpression experiment worked. However, as we know from the ISH experiments of the native developmental genes, their expression patterns appear at earlier stages (P7 and P8). Because the *gut y* stripe CRM is active much later (P10), the ectopic expression of the developmental genes may happen too late and thus fail to induce ectopic pigment. Unfortunately, we currently lack drivers that act earlier in the abdomen. Interestingly, we observed a possible enhancer-trap effect on the wings of two transgenic lines of *D. guttifera* (transformed with the *gut y* stripe CRM-hsp-*wg* cDNA construct) that produced adult flies with dark stripes along the longitudinal veins of their wings (S12 Fig). Although this phenotype was most likely not due to the influence of the *gut y* stripe CRM activity, the results suggest that the construct produces a functional Wg protein.

In order to test if the developmental genes are necessary for color pattern formation, we further attempted to knock down the mRNAs of the developmental genes *wg*, *hh*, and *dpp* by RNAi, using a ubiquitous heat shock driver that we induced during the P7 –P10 pupal stages. Unfortunately, all flies died immediately after eclosion before their color patterns could fully mature (S13 Fig). This outcome may be due to the vital roles played by these developmental genes during *Drosophila* development.

## Discussion

In this study, we show that five developmental genes (*wg*, *dpp*, *hh*, *zen*, and *abd-A*) may collectively induce the *y* gene through one CRM to orchestrate the complex abdominal spot pattern development in *D. guttifera*. We further show that *wg* expression shifted within the quinaria species group, always precisely foreshadowing the diverse abdominal spot patterns of *D. guttifera*, *D. deflecta*, *D. palustris*, and *D. subpalustris*. These results corroborate an earlier finding that Wg induces the wing spots in *D. guttifera* [6] and suggest that changes in *wg* expression may have led to the diversification of abdominal color patterns in the quinaria group.

With the exception of the *hh* gene (more precisely its downstream transcription factor Ci), the JASPAR binding site analysis suggests that the developmental genes *wg*, *dpp*, *zen*, and *abd-A* may directly activate the *gut y* spot CRM. It is worthwhile mentioning that *wg*, *dpp*, and *hh* are homologous to known proto-oncogenes in humans [28], while *zen* and *abdA* are Hox genes.

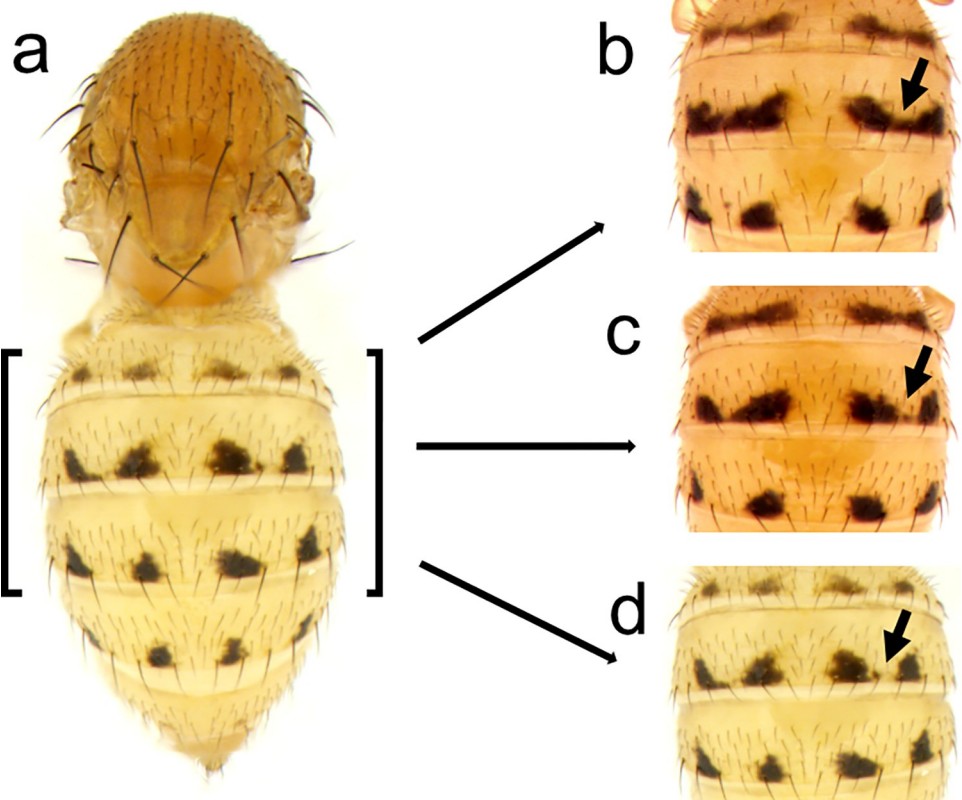

**Fig 10. Intraspecific variation of the abdominal pigment pattern of *D. falleni*.** a, Adult thorax and abdomen. b-d, The phenotypic variation of abdominal pigmentation suggests that melanin spot development may be a result of partial stripe repression (arrows). The images shown are taken from wild-collected *D. falleni* specimens from Houghton, Michigan.

The generation of the complex abdominal spot pattern in *D. guttifera* through a single CRM of the *y* gene is similar to our previous finding that the entire spot pattern on the *D. guttifera* wing is orchestrated by one CRM that responds to an elaborate expression pattern of the *wg* gene [6]. The underlying mechanism on the wing is thought to be a morphogen gradient, where Wg diffuses from a physical landmark a few cell diameters outwards to induce the *y* gene. In contrast to the *D. guttifera* wing spot pattern, the abdominal spots develop in the absence of any visible landmark structures. We speculate that the spots on the abdomen are established in a self-regulating manner, i.e., a Turing pattern [29], with the possible interplay of the morphogens Wg, Dpp, Hh, and perhaps other upstream regulators that co-regulate these developmental genes to assemble the complete pattern. Thus, the abdominal color pattern of *D. guttifera* may be regulated by multiple developmental pathways consisting of activators and repressors acting in parallel, possibly targeting pigmentation genes other than *y* as well [14, 15, 27, 30]. We acknowledge that we do not provide any functional evidence and that the regulatory relationships between the identified toolkit genes and the *y* gene are correlative.

We found that the dissection of the *gut y spot CRM* reveals a core stripe CRM that drives horizontal stripes on the posterior edges of each abdominal segment. Furthermore, the core stripe CRM appears to be conserved in a closely-related species, *D. deflecta*, which displays a similar abdominal spot pattern like *D. guttifera*. These data suggest that the abdominal spot pattern may have been shaped using an ancestral stripe element that activates *y* in stripes on the abdomen. Over time, the stripe pattern may have been partially repressed leaving isolated

spots on the abdomen. Although we did not directly identify repressor genes in this study, there is some visible evidence for the activity of pigmentation repressors found in other species of the quinaria group. As can be seen in *Drosophila falleni*'s intraspecific pigment variation (Fig 10), spots and stripes seem to be reversible pattern elements. In particular, the production of spots appears to be a result of local stripe repression, resulting in almost triangular spots.

Our multi-pathway model, involving multiple morphogens in the self-establishment of the spot patterns together with yet-to-be-discovered repressor genes fits well with the observation that the abdominal pattern variation presented by quinaria group members is largely due to modular derivations from the *D. guttifera* ground plan (Fig 3), with many species just displaying a partial pattern of what *D. guttifera* has to offer. This scenario is quite reminiscent of the modularity found in butterfly wing patterns. Because insects use similar genes for color pattern development [26, 31–34], the quinaria group may serve as a valuable model to understand insect color pattern evolution [22].

## Methods

### Molecular procedures

ISH was carried out with species-specific RNA probes, as described previously [35]. Immuno-histochemistry for the Y protein in abdomens was performed according to [12]. *D. guttifera* CRMs were identified and tested in *D. guttifera* according to [6] and in *D. melanogaster* as described previously [36]. Transgenic experiments were performed as outlined in [37]. At least five independent transgenic lines were tested for DsRed reporter expression patterns for each construct. Pupal stages were identified according to [38]. RNAi was performed using the *piggy-Bac*-based vector *Pogostick* [39].

## Supporting information

**S1 Fig. *D. guttifera* pupa stained with a *wg* probe.** This is the same pupa as in Fig 2B, but with different image manipulations. d = dorsal, m = median, l = lateral row of spots.
(TIF)

**S2 Fig. *wg* mRNA expression in four different P8 pupae of *D. guttifera*.** d = dorsal, m = median, l = lateral row of spots.
(TIF)

**S3 Fig. *wg* mRNA expression in three different P8 pupae of *D. guttifera*.**
(TIF)

**S4 Fig. *wg* mRNA expression in four different P8 pupae of *D. guttifera*.**
(TIF)

**S5 Fig.** a, *D. guttifera* P8 pupa stained with a *dpp* probe. b, *D. guttifera* P9 pupa (left) and P10 pupa (right) stained with an *abd-A* probe. c, *D. guttifera* P10 pupa stained with a *hh* probe. d, Two different *D. guttifera* P10 pupae stained with a *zen* probe. dms = dorsal midline shade, d = dorsal, m = median row of spots.
(TIF)

**S6 Fig. P7 and P8 pupal abdomens of *D. guttifera* and *D. deflecta* stained with mRNA anti-sense probes against toolkit genes, *Distal-less* (*Dll*), *Abdominal-B* (*Abd-B*), *bric-a-brac-1* (*bab-1*), *bric-a-brac-2* (*bab-2*), *engrailed* (*en*), *scalloped* (*sd*), *pannier* (*pnr*).**
(TIF)

**S7 Fig. P9 and P10 pupal abdomens of *D. guttifera* and *D. deflecta* stained with mRNA anti-sense probes against toolkit genes, *snail* (*sna*), *invected* (*inv*), *spitz* (*spi*), *Hairless* (*H*), *bric-a-brac-1* (*bab-1*), *pale* (*ple*), *spalt major* (*salm*), *Abdominal-B* (*Abd-B*), *bric-a-brac-2* (*bab-2*), *daughters against dpp* (*dad*), *engrailed* (*en*), *scalloped* (*sd*), and *Distal-less* (*Dll*).**
(TIF)

**S8 Fig.** a, Two different *D. deflecta* P9 pupae stained with a *wg* probe. b, Two different *D. deflecta* P10 pupae stained with a *hh* probe.
(TIF)

**S9 Fig. The *y* gene locus.** The horizontal bars indicate the DNA fragments of the *D. guttifera y* gene that were tested in transgenic *D. guttifera* for regulatory activity. Red: *gut y spot* CRM.
(TIF)

**S10 Fig. The full *D. guttifera y* spot CRM showing abdominal spots as well as the dorsal midline shade, shown on two different pupal abdomens from independent transgenic lines.**
(TIF)

**S11 Fig. A schematic of *gut y spot* CRM depicting the distribution of putative transcription factor binding sites along the length of the CRM.** The red solid bar is the *gut y core stripe* CRM.
(TIF)

**S12 Fig. Two transgenic *gut y* stripe CRM-hsp-*wg* cDNA lines produced adult *D. guttifera* with dark stripes along the longitudinal veins of the wings.** a, The wing of an adult *D. guttifera*, wild type. b, c, Ectopic wing pigmentation of adult *D. guttifera* expressing the *wg* cDNA construct.
(TIF)

**S13 Fig. RNAi knockdown of developmental genes in *D. guttifera*.** a, b, Knockdown of *wg* mRNA in *D. guttifera* at pupal stages P7 and P8. c, d, Knockdown of *dpp* mRNA in *D. guttifera* at pupal stage P8.
(TIF)

**S14 Fig.** ISH showing the lack of ectopic expression of the *D. melanogaster*-derived *wg* (a) and *hh* (b) genes in wildtype (non-transgenic) *D. guttifera* pupae at stage P10, acting as the negative controls for the results shown in Fig 9.
(TIF)

**S1 Table. List of selected developmental genes that were screened by ISH for their possible involvement in pigmentation on the abdomen of *D. guttifera*.**
(TIF)

**S2 Table. Numbers of putative binding sites for toolkit transcription factors for the *gut y spot* CRM by JASPAR analysis.** This table was extrapolated from the data in S9 Fig.
(TIF)

**S1 File. JASPAR analysis showing the numbers of putative binding sites for toolkit transcription factors within the *gut y spot* CRM.**
(PDF)

**S2 File. The sequence of the *gut y spot* CRM aligned with the orthologous *def y spot* CRM shows 74% sequence identity.** The *gut y core stripe* CRM is highlighted in red.
(PDF)

**S3 File.**
(DOCX)

## Acknowledgments

We thank Ryan Bensen, Abigail Meisel, Bridgette Rebbeck, and Jason Hu for technical assistance.

## Author Contributions

**Conceptualization:** Thomas Werner.

**Data curation:** Komal K. B. Raja, Mujeeb O. Shittu, Thomas Werner.

**Formal analysis:** Komal K. B. Raja, Mujeeb O. Shittu, Peter M. E. Nouhan, Tessa E. Steenwinkel, Thomas Werner.

**Funding acquisition:** Thomas Werner.

**Investigation:** Komal K. B. Raja, Mujeeb O. Shittu, Peter M. E. Nouhan, Tessa E. Steenwinkel, Evan A. Bachman, Prajakta P. Kokate, Alexander McQueeney, Elizabeth A. Mundell, Alexandri A. Armentrout, Amber Nugent.

**Methodology:** Komal K. B. Raja, Mujeeb O. Shittu, Peter M. E. Nouhan, Tessa E. Steenwinkel, Evan A. Bachman, Prajakta P. Kokate, Alexander McQueeney, Elizabeth A. Mundell, Alexandri A. Armentrout, Amber Nugent, Thomas Werner.

**Project administration:** Thomas Werner.

**Resources:** Thomas Werner.

**Supervision:** Komal K. B. Raja, Mujeeb O. Shittu, Thomas Werner.

**Validation:** Mujeeb O. Shittu, Peter M. E. Nouhan, Tessa E. Steenwinkel, Evan A. Bachman, Alexander McQueeney, Alexandri A. Armentrout, Thomas Werner.

**Visualization:** Komal K. B. Raja, Mujeeb O. Shittu, Evan A. Bachman, Thomas Werner.

**Writing – original draft:** Komal K. B. Raja, Mujeeb O. Shittu, Thomas Werner.

**Writing – review & editing:** Komal K. B. Raja, Mujeeb O. Shittu, Peter M. E. Nouhan, Tessa E. Steenwinkel, Evan A. Bachman, Prajakta P. Kokate, Alexander McQueeney, Elizabeth A. Mundell, Amber Nugent, Thomas Werner.

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
