## [Decision Letter · Decision Letter 0]

22 Sep 2022

PONE-D-22-23062The regulation of a pigmentation gene in the formation of complex color patterns in Drosophila abdomensPLOS ONE

Dear Dr. Werner,

Thank you for submitting your manuscript to PLOS ONE. After careful consideration, we feel that it has merit but does not fully meet PLOS ONE’s publication criteria as it currently stands. Therefore, we invite you to submit a revised version of the manuscript that addresses the points raised during the review process.

Please address the comments and make the suggested clarifications and changes to the text as described by the reviewer for the PLOS ONE submission. There is no need to present any additional experiments. 

We look forward to receiving your revised manuscript.

Kind regards,

Barbara Jennings

Academic Editor

PLOS ONE

Journal Requirements:

2. Please upload a new copy of Supporting Figures 11 and 13 as the detail is not clear. Please follow the link for more information:

https://blogs.plos.org/plos/2019/06/looking-good-tips-for-creating-your-plos-figures-graphics/
https://blogs.plos.org/plos/2019/06/looking-good-tips-for-creating-your-plos-figures-graphics/

Reviewers' comments:

Reviewer's Responses to Questions

**Comments to the Author**

1. Is the manuscript technically sound, and do the data support the conclusions?

Reviewer #1: Partly

2. Has the statistical analysis been performed appropriately and rigorously? 

Reviewer #1: N/A

3. Have the authors made all data underlying the findings in their manuscript fully available?

Reviewer #1: Yes

4. Is the manuscript presented in an intelligible fashion and written in standard English?

Reviewer #1: Yes

5. Review Comments to the Author

Reviewer #1: This manuscript by Raja et al. describes the analysis of the regulation of the pigmentation gene yellow in the abdomen of Drosophila guttifera and D. deflecta. It has already undergone at least one round of revisions and the authors have evidently worked hard and performed a high number of experiments. Due to the difficulties of working with non-standard model systems, experiments might not always result in the desired data, but the analysis presented here is still thorough and provides some insights. This is the first time I am reviewing this manuscript, and a few points are somewhat unclear to me.

1) Towards the end of the introduction, the authors mention divergence time between species of the quinaria group. I would also be interested in the divergence time between the focal species as well as to melanogaster since the authors later refer to the evolution between these rather distantly related species.

2) At the beginning of the results, the authors mention that they screened 110 developmental genes. Why were exactly these chosen?

3) It might be just the quality of the images in the manuscript, but I find the "spot" pattern generated by the enhancer constructs difficult to appreciate. Are there more high quality images that could supplement figures 7 and 8?

4) On p. 9, the authors write the sub-fragment displayed "horizontal" stripes. Shouldn't it be "vertical"? or even better, "parallel to the segment boundary"?

5) On the same page, the authors speculate that "the D. guttifera spots may have evolved from an ancestral stripe pattern that became partially repressed to isolate the spots". I don't think there is enough evidence to support this. This should be moved to the discussion.

6) I also find the speculation about the independent evolution of melanogaster and quinaria patterns (p. 11) a bit far-fetched and not well-supported by the data. Again, this is very speculative and would be better placed in the discussion, where it could also be elaborated on.

7) Again page 11, the authors say that the spot CRMs are not orthologous to any melanogaster sequences. How similar is the rest of the locus? Is it generally strongly diverged or are there potential CRMs that might be more conserved?

8) My main issue: Since the potential upstream factors of y show a similar expression pattern, my immediate question was, what controls their expression then? Is there a master regulator? There doesn't seem to be any speculation on how the pattern is established in the first place. What I found most puzzling throughout the manuscript was why the authors focused on the enhancer of y instead of trying to identify the enhancer(s) of wg. (I presume there are good reasons, but still.) To me, it seems much more likely that wg or a factor even more upstream evolves to lead to the development of novel patterns. wg and y could even be regulated by the same upstream factor. To be clear, I don't expect more experiments. In my opinion, this could just all be discussed a bit more.

6. PLOS authors have the option to publish the peer review history of their article (what does this mean?). If published, this will include your full peer review and any attached files.

Reviewer #1: **Yes: **Sebastian Kittelmann

---

## [Author Response · Author response to Decision Letter 0]

27 Nov 2022

We sincerely thank the reviewers for their valuable comments and their efforts to improve this manuscript. Our responses to the new comments by the reviewers are given below:

Journal Requirements:

1. Please ensure that your manuscript meets PLOS ONE's style requirements, including those for file naming. Please rename the files. The PLOS ONE style templates can be found at 

We have renamed files according to the requirements of PLOS ONE.

2. Please upload a new copy of Supporting Figures 11 and 13 as the detail is not clear. Please follow the link for more information: Please make them better

https://blogs.plos.org/plos/2019/06/looking-good-tips-for-creating-your-plos-figures-graphics/
https://blogs.plos.org/plos/2019/06/looking-good-tips-for-creating-your-plos-figures-graphics/.

We are uploading new copies of Supporting Figures 11 and 13, this time as PDF files to avoid blurring of the text.

3. Please review your reference list to ensure that it is complete and correct. If you have cited papers that have been retracted, please include the rationale for doing so in the manuscript text, or remove these references and replace them with relevant current references. Any changes to the reference list should be mentioned in the rebuttal letter that accompanies your revised manuscript. If you need to cite a retracted article, indicate the article’s retracted status in the References list and also include a citation and full reference for the retraction notice. -

We corrected the references.

RESPONSE TO THE REVIEWER’S COMMENTS

Reviewer #1: This manuscript by Raja et al. describes the analysis of the regulation of the pigmentation gene yellow in the abdomen of Drosophila guttifera and D. deflecta. It has already undergone at least one round of revisions and the authors have evidently worked hard and performed a high number of experiments. Due to the difficulties of working with non-standard model systems, experiments might not always result in the desired data, but the analysis presented here is still thorough and provides some insights. This is the first time I am reviewing this manuscript, and a few points are somewhat unclear to me.

1) Towards the end of the introduction, the authors mention divergence time between species of the quinaria group. I would also be interested in the divergence time between the focal species as well as to melanogaster since the authors later refer to the evolution between these rather distantly related species. 

We thank the reviewer for asking this important question. The divergence time between the quinaria group and D. melanogaster was dated back to 60 MYA. We added this information to the manuscript along with a reference on lines 66-70. The sentence now reads: “To better understand the development and evolution of complex color patterns, we investigated the great variety of abdominal color patterns displayed by members of the quinaria species group within the genus Drosophila, which started diverging ~10 million years ago (MYA), while the common ancestor of Drosophila melanogaster and the quinaria group was dated back to 60 MYA [8, 19-23].”

2) At the beginning of the results, the authors mention that they screened 110 developmental genes. Why were exactly these chosen? 

The 110 developmental genes were chosen because they were likely or proven to play a role in melanin formation as toolkit, immune, or terminal pigmentation genes. Many of these genes are mentioned in a book by Sean B. Carroll, which we referenced in the manuscript. Also, other genes are immune genes, as the innate immune response sometimes involves melanization events. The changes are in lines 87-90. The sentence now reads: “In order to identify putative upstream activators of y, we performed an ISH screen for 110 developmental genes (that were likely or proven to play a role in melanin formation as toolkit, immune, or terminal pigmentation genes) to detect those whose expression patterns prefigure that of the y gene (S1 Table) [24].”

3) It might be just the quality of the images in the manuscript, but I find the "spot" pattern generated by the enhancer constructs difficult to appreciate. Are there more high quality images that could supplement figures 7 and 8? 

Unfortunately, we do not have other images apart from those shown in our Supplemental Figures 7 and 8.

4) On p. 9, the authors write the sub-fragment displayed "horizontal" stripes. Shouldn't it be "vertical"? or even better, "parallel to the segment boundary"? 

We thank the reviewer for this suggestion. We have modified the sentence. The description of the expression pattern as either vertical or horizontal stripes would depend on the orientation of the fly abdomen. Therefore, we have added an additional sentence to describe the position of the fly in lines 185-189. The sentence now reads: “Unexpectedly, the 636-bp left sub-fragment displayed stripe expression parallel to the segment boundary (hereafter referred to as “horizontal stripe expression”, when viewing the fly with the head on top and the abdomen on the bottom of the picture) along the posterior edges of each abdominal segment, while the 570-bp right fragment was inactive (#1 & #2, Fig 7).”

5) On the same page, the authors speculate that "the D. guttifera spots may have evolved from an ancestral stripe pattern that became partially repressed to isolate the spots". I don't think there is enough evidence to support this. This should be moved to the discussion. 

We agree with the reviewer’s suggestion. We have moved it to the discussion (lines 337-342): It reads: “These data suggest that the abdominal spot pattern may have been shaped using an ancestral stripe element that activates y in stripes on the abdomen. Over time, the stripe pattern may have been partially repressed leaving isolated spots on the abdomen. Although we did not directly identify repressor genes in this study, there is some visible evidence for the activity of pigmentation repressors found in other species of the quinaria group.”

6) I also find the speculation about the independent evolution of melanogaster and quinaria patterns (p. 11) a bit far-fetched and not well-supported by the data. Again, this is very speculative and would be better placed in the discussion, where it could also be elaborated on. 

We agree with the reviewer’s suggestion. We deleted this sentence as it makes more sense to delete it rather than having it in the discussion.

7) Again page 11, the authors say that the spot CRMs are not orthologous to any melanogaster sequences. How similar is the rest of the locus? Is it generally strongly diverged or are there potential CRMs that might be more conserved?

We did not find similarities between the rest of the locus and D. melanogaster sequences; the sequences have strongly diverged. We found a conserved wing-body enhancer and other enhancers near the y transcription start site. We mentioned this in the manuscript in lines 211-220. The section reads: “We found the wing-body CRM approximately 4 kb upstream of the y transcription start site, whereas the CRMs that specify y expression in the head, thorax, bristles, mouth parts, legs, and trachea were found within the y intron. In addition, one 1573-bp fragment located 2 kb upstream of the y transcription start site, as shown in Fig 9a, drove reporter expression on the abdomen in horizontal stripes and resembles the reporter activity of the gut y core stripe CRM (Fig 7c). This fragment (“gut y stripe CRM”) may contain a redundant CRM, and it is possible that the spot pattern output could be due to a combinatorial activity between the two stripe CRMs and the elusive repressors.”

8) My main issue: Since the potential upstream factors of y show a similar expression pattern, my immediate question was, what controls their expression then? Is there a master regulator? There doesn't seem to be any speculation on how the pattern is established in the first place. What I found most puzzling throughout the manuscript was why the authors focused on the enhancer of y instead of trying to identify the enhancer(s) of wg. (I presume there are good reasons, but still.) To me, it seems much more likely that wg or a factor even more upstream evolves to lead to the development of novel patterns. wg and y could even be regulated by the same upstream factor. To be clear, I don't expect more experiments. In my opinion, this could just all be discussed a bit more. 

We thank the reviewer for this comment. The upstream factors of y may be regulated by multiple pathways or a master regulator, but what we found surprising is that no abdominal enhancer was found in the wg gene in a previous study by Koshikawa et al., 2015. The authors tested up to 70 Mb of the non-coding region surrounding the wg locus, but no abdominal enhancer was found. Since the upstream enhancer was not identified, we focused on the downstream y gene.

---

## [Editor Report · Decision Letter 1]

1 Dec 2022

The regulation of a pigmentation gene in the formation of complex color patterns in Drosophila abdomens

PONE-D-22-23062R1

Dear Dr. Werner,

We’re pleased to inform you that your manuscript has been judged scientifically suitable for publication and will be formally accepted for publication once it meets all outstanding technical requirements.

Kind regards,

Barbara Jennings

Academic Editor

PLOS ONE
---

## [Editor Report · Acceptance letter]

5 Dec 2022

PONE-D-22-23062R1 

The regulation of a pigmentation gene in the formation of complex color patterns in *Drosophila* abdomens 

Dear Dr. Werner:

I'm pleased to inform you that your manuscript has been deemed suitable for publication in PLOS ONE. Congratulations! Your manuscript is now with our production department. 

Kind regards, 

on behalf of

Dr. Barbara Jennings 

Academic Editor

PLOS ONE